# CO$_2$ Utilization in the Ironmaking and Steelmaking Process

**Kai Dong [1],\* and Xueliang Wang [1,2]** 

1   School of Metallurgical and Ecological Engineering, University of Science and Technology Beijing, Beijing 100083, China
2   ENFI Research Institute, China ENFI Engineering Corporation, Beijing 100038, China; xueliang1019@126.com
\*   Correspondence: dongkai@ustb.edu.cn; Tel.: +86-106-233-2122

**Abstract:** Study on the resource utilization of CO$_2$ is important for the reduction of CO$_2$ emissions to cope with global warming and bring a beneficial metallurgical effect. In this paper, research on CO$_2$ utilization in the sintering, blast furnace, converter, secondary refining, continuous casting, and smelting processes of stainless steel in recent years in China is carried out. Based on the foreign and domestic research and application status, the feasibility and metallurgical effects of CO$_2$ utilization in the ferrous metallurgy process are analyzed. New techniques are shown, such as (1) flue gas circulating sintering, (2) blowing CO$_2$ through a blast furnace tuyere and using CO$_2$ as a pulverized coal carrier gas, (3) top and bottom blowing of CO$_2$ in the converter, (4) ladle furnace and electric arc furnace bottom blowing of CO$_2$, (5) CO$_2$ as a continuous casting shielding gas, (6) CO$_2$ for stainless steel smelting, and (7) CO$_2$ circulation combustion. The prospects of CO$_2$ application in the ferrous metallurgy process are widespread, and the quantity of CO$_2$ utilization is expected to be more than 100 kg per ton of steel, although the large-scale industrial utilization of CO$_2$ emissions is just beginning. It will facilitate the progress of metallurgical technology effectively and promote the energy conservation of the metallurgical industry strongly.

**Keywords:** carbon dioxide; injection; blast furnace; converter; combustion

## 1. Introduction

Almost two tons of carbon dioxide (CO$_2$) is exhausted per ton of steel in the ferrous industry because of its energy-intensive feature [1]. Carbon dioxide utilization is beginning to attract worldwide attention, because it transmits CO$_2$ waste emissions into valuable products [2–6]. Study on the resource utilization of CO$_2$ in the ferrous metallurgy process is important for the reduction of CO$_2$ emissions to cope with global warming and bring a beneficial metallurgical effect. There are three main methods of emission reduction or utilization of CO$_2$. The first is the use of new technology or energy to reduce the use of fossil energy, the second is CO$_2$ storage technology, and the last is using CO$_2$ as a recycling resource. Currently, however, CO$_2$ emission reduction in metallurgical processes mainly relies on energy saving and waste heat utilization.

Carbon dioxide is a linear three-atom molecule, which is a weak acid, colorless and tasteless at room temperature. Its isobaric heat capacity is about 1.6 times that of nitrogen at steelmaking temperature and its infrared radiation ability is strong. CO$_2$ with CO generated from CO$_2$ can play a role in stirring in the ferrous metallurgy process. It can also play a role in controlling the temperature of the molten bath because of the carbon endothermic reaction, which protects molten steel from being oxidized and dilutes oxidants in the combustion. In the realization of CO$_2$ emission reduction, it is

possible to save energy and reduce costs, and at the same time, dephosphorize and remove inclusions in steel.

$CO_2$ is a weaker oxidizing agent compared with $O_2$. The reactions of $CO_2$ oxidizing carbon, iron, silicon, or manganese in a molten bath may occur at steelmaking temperature, and the reactions are endothermic or slightly exothermic reactions, respectively. Therefore, it is possible to control the temperature and atmosphere by adopting $CO_2$ in the steelmaking process, hence realizing the aims of (1) reducing dust generation by reducing the temperature in the fire zone, (2) purifying the liquid steel by promoting dephosphorization, (3) minimizing the loss of valuable metals by low oxidation, (4) saving energy by increasing gas recovery, and (5) reducing the total consumption. The current $CO_2$ applications in the ferrous metallurgy process are shown in Figure 1.

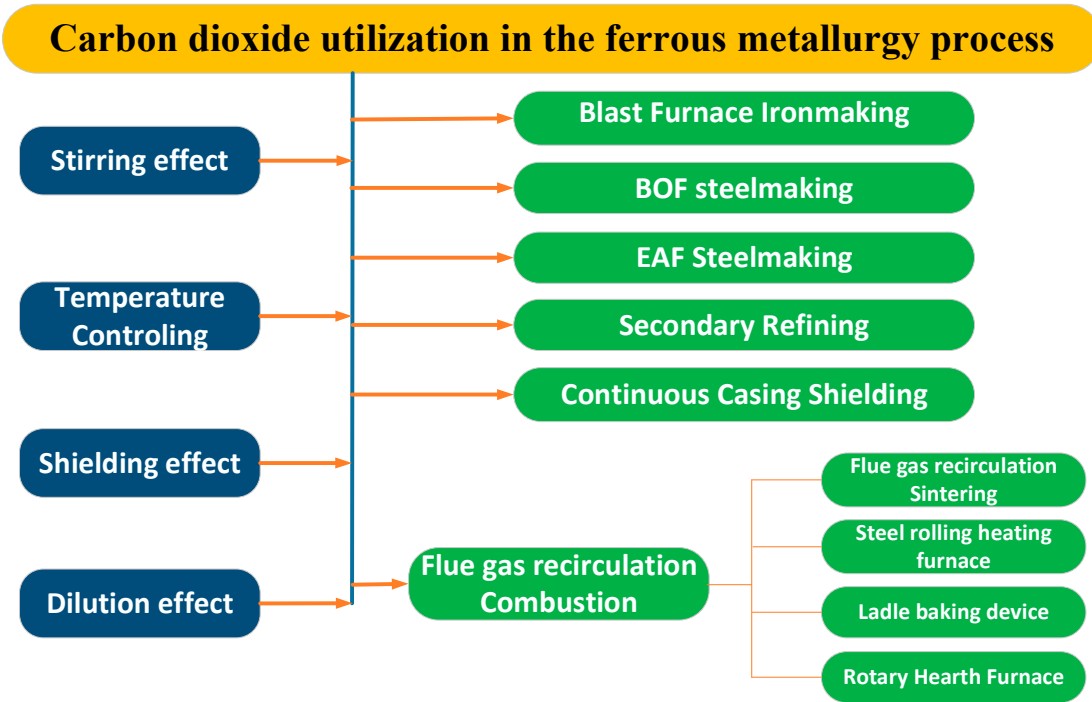

**Figure 1.** Carbon dioxide utilization in the ferrous metallurgy process.

## 2. Application in the Steelmaking Process.

### 2.1. Top-Blowing Carbon Dioxide in Converter

As Table 1 shows, compared with pure oxygen, the chemical heat release decreases when $CO_2$ is used in the steelmaking process as an oxidizing agent due to the endothermic or micro exothermic reaction of $CO_2$ in the molten bath, which may mean a reduced capacity for scrap melting. Therefore, when a certain proportion of $CO_2$ is blown into the converter in the dephosphorization process, the temperature is controlled, and suitable thermodynamic conditions for the dephosphorization reaction are created. At the same time, the reaction produces more stirring gas, which is conducive to strengthen bath stirring and creates favorable dynamic conditions for the dephosphorization reaction too.

Since 2004, Zhu Rong et al. [7–14] have carried out basic exploration research and industrial tests involving $CO_2$ utilization in steelmaking. After researching this area for a decade, they applied $CO_2$ in the top-blowing of the Basic Oxygen Furnace (BOF) and researched the $CO_2$ injection technology involved in top-blowing in the converter. When blowing $CO_2$ in the converter, the dust is reduced by 19.13%, the total Fe in the dust decreases by 12.98%, and the slag iron loss is reduced by 3.10%. As an agitation effect achieved from the temperature controlling improvement, the dephosphorization rate increases by 6.12%, and the nitrogen content of the molten steel reduces too, which leads to a higher

quality of the molten steel. According to some test results, about 80~90% of $CO_2$ blown into the BOF participates in the chemical reaction at 1600 °C [15].

**Table 1.** Chemical reaction thermodynamic data of $CO_2$ with elements in the molten iron.

| Elements | Chemical Reactions | $\Delta G^\theta$ (J/mol) | $\Delta G^\theta$ (kJ/mol) 1773 K | $\Delta H$ (kJ/mol) 298 K |
|---|---|---|---|---|
| C | $1/2O_2 + [C] = CO(g)$ | $-140,580 - 42.09\,T$ | $-215.21$ | $-139.70$ |
| | $O_2 + [C] = CO_2(g)$ | $-419,050 + 42.34\,T$ | $-343.98$ | $-393.52$ |
| | $CO_2(g) + [C] = 2CO(g)$ | $137,890 - 126.52\,T$ | $-86.43$ | $172.52$ |
| Fe | $1/2O_2(g) + Fe(l) = (FeO)$ | $-229,490 + 43.81\,T$ | $-151.81$ | $-272.04$ |
| | $CO_2(g) + Fe(l) = (FeO) + CO(g)$ | $48,980 - 40.62\,T$ | $-23.04$ | $40.37$ |
| Si | $O_2 + [Si] = (SiO_2)$ | $-804,880 + 210.04\,T$ | $-432.48$ | $-910.36$ |
| | $2CO_2(g) + [Si] = (SiO_2) + 2CO(g)$ | $-247,940 + 41.18\,T$ | $-174.93$ | $-344.36$ |
| Mn | $1/2O_2 + [Mn] = (MnO)$ | $-412,230 + 126.94\,T$ | $-187.17$ | $-384.93$ |
| | $CO_2(g) + [Mn] = (MnO) + CO(g)$ | $-133,760 + 42.51\,T$ | $-58.39$ | $-101.91$ |

Top-blowing $CO_2$ in the converter is a major innovation in China and has been applied in Shougang Jingtang Company, who have achieved good results. $CO_2$ sources are wide in the iron and steel enterprises, providing a convenient condition for the converter blowing $CO_2$.

### 2.2. Bottom-Blowing Carbon Dioxide in the BOF

In the 1970s, the scholars began to study bottom-blowing $CO_2$ in the converter steelmaking process and found [9] that $CO_2$ can participate in the bath reaction, and its bottom-blowing agitation ability is stronger than that of Ar and $N_2$, when compared with blowing $N_2$/Ar from the bottom, which is an easy way to make [N] increase, and blowing $O_2$/$C_xH_y$ from the bottom, which is an easy way to make [H] increase. $CO_2$ is an effective alternative to high-cost Ar and potentially harmful $N_2$ [16–18].

In the 1990s, researchers from AnGang Company [19] studied the bottom blowing of $CO_2$ in the top and bottom blowing converter and found that $CO_2$ can be used for bottom blowing of the top-bottom blowing converter. To prevent the strong cooling effect of the bottom blowing $CO_2$ gas from the nozzle being clogged, some oxygen was mixed in the bottom blowing gas. Unfortunately, the use of this technique was stopped due to issues with the blowing brick's life.

Recent studies have found that by bottom blowing $CO_2$, the slag iron loss can be reduced, the bath stirring can be strengthened, and the dephosphorization rate can be improved. In 2009, industrial experiment of bottom-blowing $CO_2$ on the 30 t converter was conducted in Fujian Sanming Steel company (Sanming, Fujian, China). The results showed that the converter with $CO_2$ bottom blowing is feasible, with no obvious erosion of the hearth [20].

### 2.3. Bottom-Blowing Stirring in the Ladle Furnace

Previously, to avoid re-oxidation and hydrogen and nitrogen absorption, the bottom blowing of $CO_2$ instead of Ar for stirring in a ladle was studied. The stirring mechanism of $CO_2$ bottom blowing during the ladle furnace (LF) refining process was studied [21]. As an exploratory industrial experiment, different proportions of $CO_2$ and Ar gas mixtures were bottom blown in a 75 t LF. The results showed that the stirring was reinforced when blowing $CO_2$, and the desulfurization rate was increased from 49.7% to 65.1%, and the average slag (FeO) content was less than 0.5%, which meets the oxidizing slag requirements. Though the type, the morphology, and the composition of the inclusions in molten steel changed little, the average number of inclusions per analyzed area decreased, which means the cleanliness of the liquid steel improved. The tests showed that the LF furnace can use $CO_2$ gas for refining.

## 2.4. Bottom-Blowing Stirring in the Electric Arc Furnace

Since the emergence of Electric Arc Furnace (EAF) bottom blowing technology in the 1980s, scholars have studied and found that the EAF bottom blowing technology can improve the bath agitation ability, promote the inter-slag reaction, uniform bath temperature, and composition, and improve the alloy yield. It is of great significance to improve the furnace dynamics.

Industrial experiments [21] of bottom-blowing $CO_2$ in a 65 t Consteel EAF verified that bottom-blowing $CO_2$ instead of Ar is feasible. The studies showed that compared with the conventional bottom-blown Ar process, bottom-blowing $CO_2$ increases the end [C] content and oxidizes a small amount of [Cr], but the contents of [Mn], [Mo], [O] and [N] are not influenced. This method can also enhance the bath stirring, raise the basicity, and reduce the slag (FeO) content. It provides a suitable kinetic and thermodynamic condition for EAF desulfurization and dephosphorization, with the desulfurization degree increasing by 7%.

## 2.5. Shielding Gas in Continuous Casting Process

In 1989, reference [22] stated that a US company applied $CO_2$ instead of Ar to protect the injection flow when casting special steel rod. Benefiting from a higher density, $CO_2$ will fall in parallel with the injection flow through the upper portion of the upper casing or spiral holes, maintaining the positive pressure of the stream around to prevent air suction. It shows a great effect in cutting off the liquid steel from the air and preventing the oxidation of the molten steel.

To solve problems of using $CO_2$ with specific process issues, our team conducted experiments and found that when using $CO_2$ instead of Ar in submerged nozzle seal protection, the [N] content increased. When bottom blowing Ar for 40Cr steel and 45 steel, the [N] content increased by 10.4% and 53.6%, respectively. While bottom blowing $CO_2$, the [N] content increased by 17.6% and 54.4%, respectively. When observing $CO_2$-protected steel, the [O] content was shown to be decreased. $CO_2$ can play a protective role in casting, which can reduce secondary oxidation.

The amount of $CO_2$ emission utilization in the BOF steelmaking process will be greater than 40 kg/ton of steel. Unfortunately, reports of sustainable utilization of $CO_2$ emissions in the steelmaking industry are almost non-existent, because the industrial supply of $CO_2$ gas has not yet been established yet.

## 3. Application in Sintering and Blast Furnace

### 3.1. Flue Gas Recirculation (FGR) Sintering

The main combustion produce gas in sintering is $CO_2$. Typical Flue Gas Recirculation (FGR, Figure 2) sintering [23] includes the waste gas regional recirculation process developed by Nippon Steel [24], and similar technologies have been developed now, such as the Exhaust Gas Recirculation (EGR) sintering technology developed by Hata [25], the Emission Optimized Sintering (EOS) process developed by Corus Ijmuiden in Netherlands [26], the LEEP (low emission and energy optimized sintering process) developed by HKM [24] and the EPOSINT (environmental process optimized sintering) process developed by Siemens VAI [27–29].

The existing recycling technology still has the following disadvantages [23,24]:(1) In the Nippon Steel recycling process, only high oxygen flue gas is circulated. The flue gas emission reduction rate is relatively low, about 28%. The circulation process is complex, and it is difficult for modifying a sintering machine. (2) EOS technology has not considered the characteristics of sintering flue gas emissions, and the effect of dealing with different components in the flue gas is not the best. (3) For the LEEP technology, the heat of the high temperature flue gas is not fully utilized. The rear part of the flue gas has a high content of $SO_2$, which results in the sintering ore [S] content increasing. (4) The EPOSINT process only makes a high sulfur gas cycle, and the reduction in emission rates is small, only 28~25%, and the sintering ore [S] content increases too. In addition, the high temperature flue gas is not circulating, and the energy saving rate and the dioxin emission reduction rate are both low.

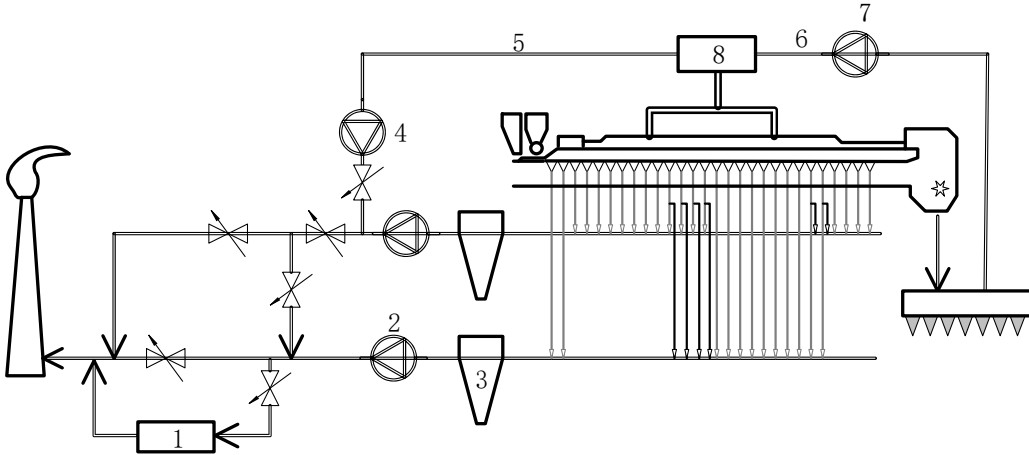

1-Desulfurization system, 2-Main exhaust fan, 3-Dust catcher, 4-Circulating air blower, 5-Large flue circulating flue gas, 6-Air cooling hot exhaust gas, 7-Air cooling heat recovery fan, 8-Flue gas mixing chamber.

**Figure 2.** Flow diagram of flue gas circulation sintering process.

### 3.2. $CO_2$ Injection Through the Blast Furnace Tuyere

In 2010, Fu Zhengxue [30] et al. developed a method of injecting carbon dioxide into a Blast Furnace (BF). Blowing $CO_2$ or the exhaust gas containing $CO_2$ into BF can effectively solve some problems, such as resource saving, $CO_2$ emissions reduction, and environmental pollution elimination. The technical scheme is as follows: Firstly, $CO_2$ or waste gas containing $CO_2$ is blown into BF cold air pipes. After heating in the hot blast stove, $CO_2$ is blown or sprayed into the BF tuyere zone by the hot air pipeline. Blown oxygen and $CO_2$ react with the burning carbon in the tuyere zone, and then CO is generated. Because of $CO_2$ absorbing part of the exhausted heat of oxygen, a higher oxygen enrichment can be achieved, and the CO generated is the smelting reduction agent, so the reductant rate of the Blast Furnace is increased, and the production efficiency of BF will be improved. However, the technology has not been applied industrially.

### 3.3. Carrier Gas of Pulverized Coal Injection

In 2011, Chinese researchers [31] invented a method to use $CO_2$ as the transmission medium of pulverized coal injected into BF (Figure 3). Mixed coal powder was used in BF tuyere to replace coke, provide heat, and act as a reducing agent.

Pulverized coal in the tuyere zone reacts not only with the enriched $CO_2$, but also with the oxygen from hot air. It is necessary to adjust the amount of pulverized coal injection and the oxygen enrichment level to achieve the best ratio and promote complete combustion of the pulverized coal in front of the tuyere. Due to the use of $CO_2$ instead of compressed air or nitrogen as the transmission medium, the total amount of exhaust gas through BF should be reduced, which will significantly reduce the nitrogen content and increase the amount of $CO_2$ + CO. Therefore, the purity and calorific value of BF gas increase.

So far, there has been no industrial application reported for the above two patents. $CO_2$ injection through a Blast Furnace tuyere creates a new route of $CO_2$ utilization. The amount of $CO_2$ utilization in the BF process will be greater than 50 kg/ton of iron, but the utilization of $CO_2$ emissions in the BF process is still being researched.

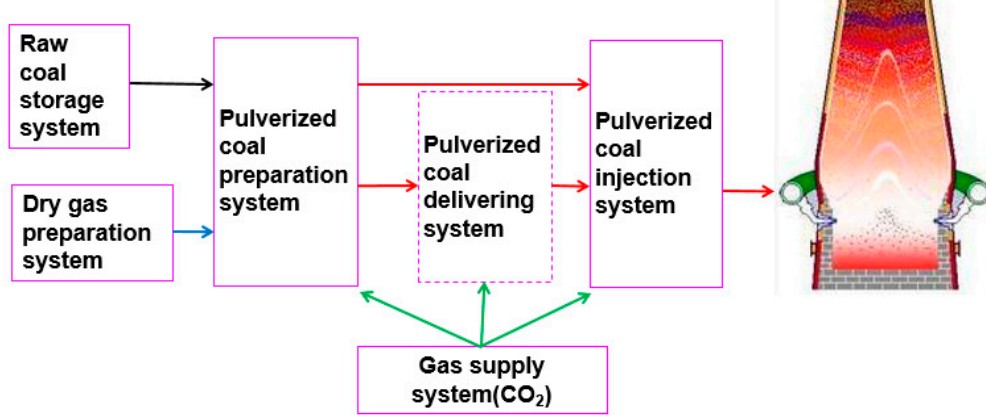

**Figure 3.** Flow diagram of pulverized coal injection of BF.

## 4. Applications in Other Ferrous Metallurgy Processes

### 4.1. Application in Smelting Stainless Steel

In 2011, Anshan Iron and Steel Co., Ltd. [32] invented a smelting method of AOD by blowing $CO_2$ to produce stainless steel, which is mainly about injecting $CO_2$ into molten steel to enhance the decarburization. This suits smelting steel whose carbon content ranges from 0.001% to 0.3%. $CO_2$ injected into the molten steel not only can decarburize, but can also enhance the bath stirring, and it can promote the oxygen reaction and cool the oxygen lance, whose life is improved by 20%.

Our team studied the mechanism of Cr retention and decarburization when injecting $CO_2$ to AOD from the aspects of thermodynamics and kinetics. In the laboratory, tube furnace experiments found that the Cr retention and decarburization effects of $CO_2$ are very good, and the carbon content can reach 0.5% with little chromium oxidation occurring [33]. When the $O_2$ proportion of smelting gas was increased, large amounts of chromium oxidation occurred, and the decarburization was relatively reduced. An analysis of the changes in materials and energy balance found that it could meet the requirements for refining the temperature when the $CO_2$ injection ratio was less than 9.13%. As the proportion of $CO_2$ increased within this range, the AOD furnace surplus heat reduced, and the CO proportion of furnace gas increased.

### 4.2. $CO_2$ Circulation Combustion

$CO_2$ circulation combustion technology uses recycled flue gas to replace the nitrogen in the air (Figure 4), which can reduce nitric oxide emissions and improve the thermal efficiency of combustion [34]. Its application in steel rolling heating furnaces and pit furnaces can reduce fuel consumption, shorten the heating time, and reduce emissions.

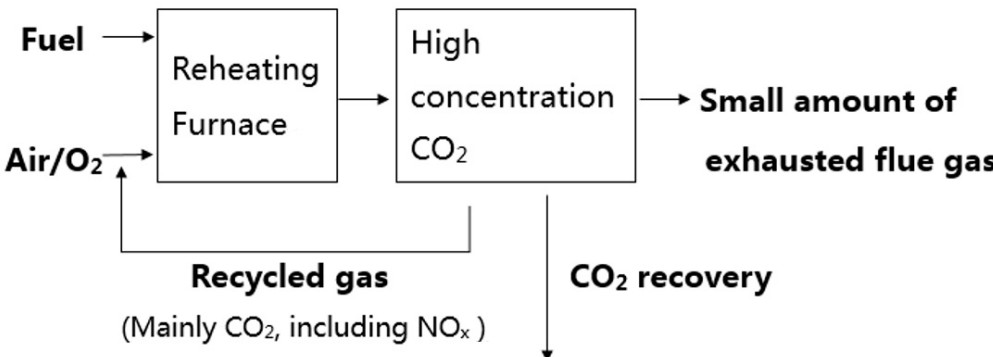

**Figure 4.** Flow diagram of flue gas circulation combustion technology.

In the 1920s, the Linde Group [35] tried applying flameless combustion for heating in the ferrous industry, reducing fuel consumption and emissions of $NO_x$ and $CO_2$ through the development of a special burner to achieve a special method of combustion. In addition, the Swedish Hofors works Ovako also used the technology in the heating furnace, finding that the production increased by 30–50% and the fuel reduced by 30–45% with more uniform heating and reductions in $CO_2$ and $NO_x$ emissions.

## 5. Conclusions

There are many methods of $CO_2$ utilization in the ferrous metallurgy process, such as directly being blown in the BF and Converter, serving as a carrier gas for coal injection in BF, use as a shielding gas and mixing gas in refinement, the continuous casting process and stainless steelmaking, and use in circulation in the exhaust gas of the steel rolling heating furnace. All of these uses reflect the special role that $CO_2$ plays in the ferrous metallurgy process.

In China, policies of $CO_2$ utilization and emission reduction are actively carried out. The historical carbon emissions are being investigated to build a unified nationwide carbon emissions trading system. Meanwhile, some Chinese steel plants have carried out procedures involving $CO_2$ utilization. The technology of top and bottom blowing $CO_2$ in the 300 t converter has been applied, which achieved good results.

With the continuous improvement and further expansion of $CO_2$ application in the ferrous metallurgy process, carbon dioxide usage in the ferrous metallurgy process is expected to be more than 100 kg per ton of steel. At present, the annual steel output of China is about 800 million tons, and the annual amount of recycled $CO_2$ utilization is around 80 million tons in metallurgical processes, which could effectively facilitate the progress of metallurgical technology, strongly promoting energy conservation in the metallurgical industry. It meets the need for sustainable development in China.

**Author Contributions:** Writing-Original Draft Preparation, K.D. and X.W; Writing-Review & Editing, K.D.; Visualization, K.D. and X.W; Supervision, X.W.; Project Administration, K.D.

**Conflicts of Interest:** The authors declare no conflict of interest.

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
