# Peer review of "CO2 Utilization in the Ironmaking and Steelmaking Process"

_metals, doi:10.3390/met9030273_

Reviewer 1 Report

The manuscript discusses potentially interesting uses of carbon dioxide in ironmaking and steelmaking. However, it is my opinion that several aspects of the manuscript would need to be improved before this work would be publishable:

1. Extensive language editing is needed.

2. Several inexact statements should be quantified better. Examples include the following:

a) Give the actual CO2 emissions per ton of steel produced (rather than referring to "large amounts" in line 26).

b) The heat capacity of CO2 is 1.3 times that of N2 at room temperature, and 1.67 times above 1100 K – not "about 1.6 times higher" (line 36).

c) In steelmaking (section 2.1), carbon dioxide could in principle replace some oxygen, but at the cost of reduced capacity to melt scrap. This trade-off should be quantified.

d) It is stated that the "main component [of] flue gas in sintering is CO2". This is not the case for conventional sintering (where nitrogen is the main component).

e) If CO2 is injected into the blast furnace (sections 3.2 and 3.3), it would inevitably increase the reductant rate, because the top gas is a mixture of CO and CO2. Injecting CO2 into a blast furnace would thus INCREASE the overall CO2 intensity of ironmaking, and thus it appears that there is no reason to use CO2 as injectant. This should be discussed.

f) In line 197, it is stated that if oxygen were included in the gas used in stainless steelmaking, "the decarburization effect would be reduced". This is not correct.

3. It is not at all clear how the stated properties of carbon dioxide (quoted in lines 35-37) are relevant to its use as a stirring gas.

4. A fundamental problem with use of CO2 in ironmaking and steelmaking is that it is an oxidizing gas, which would appear to preclude its use in ironmaking and secondary metallurgy. This should be discussed in more detail, giving attention to both kinetic and thermodynamic aspects. Bland statements such as that carbon dioxide protects "molten steel from being oxidized" (line 39) needs much better qualification.
The work of Sun and Jahanshahi (see, for example, Metallurgical and Materials Transactions B, vol. 31, pp. 937-945, 2000) on oxidation by CO2 should be considered.

5. Reference is made to Table 1, but it was not included in the manuscript.

Author Response

Dear Peer reviewer

I have revised the manuscript according to your suggestions, the detailed responses are shown as follows.

1. Extensive language editing is needed.

Response:

I have improved my English editing, and I’m asking help from a language editing service, but which needs some time.

 2. Several inexact statements should be quantified better. Examples include the following:

a) Give the actual CO2 emissions per ton of steel produced (rather than referring to "large amounts" in line 26).

Response:

Almost 2 tons of carbon dioxide (CO2) was exhausted per ton of steel in the ferrous industry.

 b) The heat capacity of CO2 is 1.3 times that of N2 at room temperature, and 1.67 times above 1100 K – not "about 1.6 times higher" (line 36).

Response:

Its isobaric heat capacity is about 1.6 times that of nitrogen at steelmaking temperature.

 c) In steelmaking (section 2.1), carbon dioxide could in principle replace some oxygen, but at the cost of reduced capacity to melt scrap. This trade-off should be quantified.

Response:

Exothermic data are in table1, and the missing table has been added.

 d) It is stated that the "main component [of] flue gas in sintering is CO2". This is not the case for conventional sintering (where nitrogen is the main component).

Response:

The main combustion produce gas in sintering is CO2.

 e) If CO2 is injected into the blast furnace (sections 3.2 and 3.3), it would inevitably increase the reductant rate, because the top gas is a mixture of CO and CO2. Injecting CO2 into a blast furnace would thus INCREASE the overall CO2 intensity of ironmaking, and thus it appears that there is no reason to use CO2 as injectant. This should be discussed.

Response:

Because of CO2 absorbing part exhausted heat of oxygen, a higher oxygen enrichment can be achieved, and CO generated is the smelting reduction agent, the production efficiency of BF will be improved.

f) In line 197, it is stated that if oxygen were included in the gas used in stainless steelmaking, "the decarburization effect would be reduced". This is not correct.

Response:

When O2 proportion of smelting gas is increased, large amounts of chromium oxidation occurred, and the decarburization was relatively reduced

 3. It is not at all clear how the stated properties of carbon dioxide (quoted in lines 35-37) are relevant to its use as a stirring gas.

Response:

CO2 with CO generated from CO2 can play a role in stirring in the ferrous metallurgy process.

4. A fundamental problem with use of CO2 in ironmaking and steelmaking is that it is an oxidizing gas, which would appear to preclude its use in ironmaking and secondary metallurgy. This should be discussed in more detail, giving attention to both kinetic and thermodynamic aspects. Bland statements such as that carbon dioxide protects "molten steel from being oxidized" (line 39) needs much better qualification.
The work of Sun and Jahanshahi (see, for example, Metallurgical and Materials Transactions B, vol. 31, pp. 937-945, 2000) on oxidation by CO2 should be considered.

Response:

In this manuscript, different methods of CO2 utilization in ironmaking and steelmaking basing on the different stated properties are introduced, so the principles are introduced following with the methods.

 5. Reference is made to Table 1, but it was not included in the manuscript.

Response:

The missing table has been added.

Reviewer 2 Report

The subject of the manuscript is interesting and current. The authors describe the use of CO2 in steel making metallurgy. However, the manuscript has flaws that need revising before the manuscript if acceptable for publishing. Overall, the level of English language in the manuscript is very hard to understand. I strongly recommend revision of the English in the manuscript and using a native English speaker in the revision process.

Comments:

 Abstract

1)      line 13-14: The authors write According to… Should the phrase rather be based on… Now it sounds like the paper analysed things from some report instead of authors critical evaluations.

2)      It would be interesting to know what is the current utilization of CO2 in steelmaking.

 Introduction

1)      line 29: Replace ‘significant’ with important. In my opinion, the word significant is related to statistics more than anything.

2)      line 30: Are you saying that utilization of CO2 in steelmaking copes with global warming alone

3)      line 31: How do you propose to develop new energy?

4)      line 41: Be more specific how the steel quality improves. Quality is subjective and depends on what is the target.

5)      line 42: It is possible… This phrase is very hard to understand.

6)      lines 44-48: Therefore, it is…. This phrase is vague an unspecified. Probably all steelmaking tries to accomplish these things. Could you give some numbers how exactly CO2 use changes all these points?

 Application in steelmaking process

1)      Table 1 is missing

2)      lines 53-54: how can the thermal effect be relatively decreased?

3)      line 57: Do you mean ‘favorable’ or ‘suitable’ thermodynamic conditions? I don’t think that thermodynamic conditions are either good or bad.

4)      line 68: Again, quality depends on the target and it is unclear here what is the target. Differents steels have different notions of good quality.

5)      lines 69-70: To which studies are you referring to in this phrase?

6)      lines 132-133: It would be interesting to know what is the current utilization of CO2 emissions in steelmaking process

 Applications in sintering and blast furnace

1)      lines 136-141: Are all these technologies in one FGR?

2)      lines 161-163: Please be more specific and give for example numbers on how exactly and to what extent are these problems solved

3)      lines 183-184: It would be interesting to know the current utilization too

4)      Reference in Fig.5 is missing [Nyby37]

 Author Response

Dear Peer reviewer

I have revised the manuscript according to your suggestions, and I’m asking help from a language editing service, but which needs some more time. The detailed responses are shown as follows.

 Abstract

1)      line 13-14: The authors write According to… Should the phrase rather be based on… Now it sounds like the paper analysed things from some report instead of authors critical evaluations.

Response:

‘According to’ is replaced by ‘based on’.

 2)      It would be interesting to know what is the current utilization of CO2 in steelmaking.

Response:

The large-scale industrial utilization of CO2 emissions is just beginning.

 Introduction

1)      line 29: Replace ‘significant’ with important. In my opinion, the word significant is related to statistics more than anything.

Response:

‘significant’ is replaced by “important”.

2)      line 30: Are you saying that utilization of CO2 in steelmaking copes with global warming alone

Response:

Study on resource utilization of CO2 in the ferrous metallurgy process is significant important for the reduction of CO2 emissions and, the coping with global warming and bringing beneficial metallurgical effect.

 3)      line 31: How do you propose to develop new energy?

Response:

there are three main ways to the emission reduction and utilization of CO2. The first is to develop new technology and energy and to reduce the use of fossil energy, the second is to develop CO2 storage technology, and the last is to use CO2 as a recycling resource.

 4)      line 41: Be more specific how the steel quality improves. Quality is subjective and depends on what is the target.

Response:

It is possible to save energy and reduce costs, and at the same time, dephosphorizate and remove inclusions in steel

 5)      line 42: It is possible… This phrase is very hard to understand.

Response:

The reactions of CO2 oxidizing with carbon, iron, silicon and or manganese in molten bath may occur at steelmaking temperature.

 6)      lines 44-48: Therefore, it is…. This phrase is vague an unspecified. Probably all steelmaking tries to accomplish these things. Could you give some numbers how exactly CO2 use changes all these points?

Response:

1) reducing the dust generation by reducing the temperature in fire zone, 2) purifying the liquid steel by promoting dephosphorization, 3) minimizing the loss of valuable metals by low oxidation, 4) saving the energy by increasing gas recovery

 Application in steelmaking process

1)      Table 1 is missing

Response:

The missing table has been added.

 2)      lines 53-54: how can the thermal effect be relatively decreased?

Response:

Exothermic data are in table1, and the missing table has been added.

 3)      line 57: Do you mean ‘favorable’ or ‘suitable’ thermodynamic conditions? I don’t think that thermodynamic conditions are either good or bad.

Response:

 ‘good’ is replaced by ‘suitable’.

 4)      line 68: Again, quality depends on the target and it is unclear here what is the target. Differents steels have different notions of good quality.

Response:

As the agitation effect and the temperature controlling is improved, the dephosphorization rate is increased by 6.12%, reducing the nitrogen content of the molten steel, which mean higher quality of the molten steel.

 5)      lines 69-70: To which studies are you referring to in this phrase?

Response:

A new reference [15] is added

 6)      lines 132-133: It would be interesting to know what is the current utilization of CO2 emissions in steelmaking process

Response:

Unfortunately, the sustainable utilization of CO2 emissions in steelmaking industrial is nearly blank, because the industrial supply of CO2 gas has not yet been established yet.

 Applications in sintering and blast furnace

1)      lines 136-141: Are all these technologies in one FGR?

Response:

Typical Flue Gas Recirculation sintering is developed by Nippon Steel, and similar technologies have been developed, Such as EGR, EOS, LEEP and EPOSINT.

 2)      lines 161-163: Please be more specific and give for example numbers on how exactly and to what extent are these problems solved

Response:

Oxygen and CO2 blown all reacts with the burning carbon in the tuyere zone, and then CO generates. Because of CO2 absorbing part exhausted heat of oxygen, a higher oxygen enrichment can be achieved, and CO generated is the smelting reduction agent of BF, the production efficiency of BF will be improved.

 3)      lines 183-184: It would be interesting to know the current utilization too

Response:

the utilization of CO2 emissions in BF process is still in the research period

 4)      Reference in Fig.5 is missing [Nyby37]

Response:

Reference in Fig.5 should be [34] now

Round  2

Reviewer 1 Report

Extensive language editing is still required.

The fifth and last reactions in Table 1 are not balanced.

Explain what the round and square brackets in Table 1 indicate.

What evidence is there for the statement that CO2 can dephosphorize steel and remove inclusions (second paragraph of Introduction)?

In section 2.1, explicit mention must be made of the reduced capacity for scrap melting if CO2 is blown into a steelmaking converter.

In section 2.3, what does it mean when it is stated that the "equivalent density of the inclusion was decreased"?

In section 2.5, the effect of CO2 on the nitrogen concentration in steel should be explained.

In section 3.2, it is essential to mention that CO2 injection will increase the reductant rate of the blast furnace, increasing the CO2 emissions from blast furnace ironmaking.

Author Response

Dear Peer reviewer

I have revised the manuscript according to your suggestions, the detailed responses are shown as follows.

Extensive language editing is still required.

Response:

I request English editing help from MDPI.

 The fifth and last reactions in Table 1 are not balanced.

Response:

I’ m sorry. Because the typesetting mistakes, the three “+CO(g)” were blocked, and I didn’t find the mistakes. The layout of the table has been revised now.

 Explain what the round and square brackets in Table 1 indicate.

Response:

In metallurgical physical chemistry, round brackets mean the content in molten slag, and square brackets mean the content in molten metal.

 What evidence is there for the statement that CO2 can dephosphorize steel and remove inclusions (second paragraph of Introduction)?

Response:

dephosphorization by CO2 is detailed introduced in section 2.1, and removing inclusions by CO2 is introduced in section 2.3 and in the first paragraph of section 2.5.

 In section 2.1, explicit mention must be made of the reduced capacity for scrap melting if CO2 is blown into a steelmaking converter.

Response:

compared with pure oxygen, the chemical heat release is relatively decreased when CO2 is used in the steelmaking process as oxidizing agent, due to the endothermic or micro exothermic reaction of CO2 in the molten bath, which may mean the reduced capacity for scrap melting.

By reducing the slag quantity, because of the higher dephosphorization rate, the scrap ratio is no significant changed in the industrial application

 In section 2.3, what does it mean when it is stated that the "equivalent density of the inclusion was decreased"?

Response:

The average density of the inclusion was decreased

 In section 2.5, the effect of CO2 on the nitrogen concentration in steel should be explained.

Response:

Benefiting from a higher density, CO2 shows great effect in cutting off the liquid steel, and limits the nitrogen absorption from air, as introduced in the first paragraph of section 2.5.

 In section 3.2, it is essential to mention that CO2 injection will increase the reductant rate of the blast furnace, increasing the CO2 emissions from blast furnace ironmaking.

Response:

Because of CO2 absorbing part exhausted heat of oxygen, a higher oxygen enrichment can be achieved, so the reductant rate of the blast furnace is increased, and the production efficiency of BF will be improved.

Oxygen and CO2 blown all react with the burning carbon and generate into CO firstly; CO gradual reacts with Iron ore and forms CO2 secondly. The second reaction is limited by height of the iron ore, quantity of heat and so on. In general, the CO2 emissions from blast furnace ironmaking decreases, and calorific value of blast furnace gas increases, which can substitute for other fuels.

Reviewer 2 Report

I think that the authors have increased the quality of the manuscript. Before publishing, the language of the manuscript should be carefully checked as currently some phrases can be misleading. The authors claim that the manuscript is going through a professional language check. In addition, I have some minor comments on the content:

 You should revise the title for example as ‘CO2 Utilization in the Iron- and Steelmaking Processes’

 2 Application in steelmaking process

2.1Top-blowing carbon dioxide in Converter

I suggest you write ‘…oxygen, the chemical heat release decreases…’ rather than ‘…oxygen, the chemical heat release is relatively decreased…’

 4 Application in other ferrous metallurgy process

4.1 Application insmelting stainless steel

You write ‘…Our teamstudied the mechanism of…’ Could you provide some kind of reference for this experimental work?

 5 Conclusions

You state ‘…carbon dioxide usage in the ferrous metallurgy process is expected to be 100~200kg per ton steel.’ Then, in the next phrase you state that CO2 utilization is 80 million tonnes, which equals the 200 kg per ton usage. However, in the abstract you state that the expected use of CO2 is 100 kg per ton of steel. Moreover, you state in the manuscript that the CO2 use is an emerging technology. I think you must be more consistent with amount of CO2 used (100 or 200 or both or within that range) and clearly distinguish between current and expected use.

 Author Response

Dear Peer reviewer

I have revised the manuscript according to your suggestions, the detailed responses are shown as follows.

I think that the authors have increased the quality of the manuscript. Before publishing, the language of the manuscript should be carefully checked as currently some phrases can be misleading. The authors claim that the manuscript is going through a professional language check.

Response:

I request English editing help from MDPI.

You should revise the title for example as ‘CO2 Utilization in the Iron- and Steelmaking Processes’

Response:

In Metallurgical Industry, “Ironmaking and Steelmaking” is a generic name.

 2 Application in steelmaking process

2.1Top-blowing carbon dioxide in Converter

I suggest you write ‘…oxygen, the chemical heat release decreases…’ rather than ‘…oxygen, the chemical heat release is relatively decreased…’

Response:

the chemical heat release decreases, when CO2 is used in the steelmaking process as oxidizing agent

4 Application in other ferrous metallurgy process

4.1 Application insmelting stainless steel

You write ‘…Our teamstudied the mechanism of…’ Could you provide some kind of reference for this experimental work?

Response:

A new reference [33] is added.

 5 Conclusions

You state ‘…carbon dioxide usage in the ferrous metallurgy process is expected to be 100~200kg per ton steel.’ Then, in the next phrase you state that CO2 utilization is 80 million tonnes, which equals the 200 kg per ton usage. However, in the abstract you state that the expected use of CO2 is 100 kg per ton of steel. Moreover, you state in the manuscript that the CO2 use is an emerging technology. I think you must be more consistent with amount of CO2 used (100 or 200 or both or within that range) and clearly distinguish between current and expected use.

Response:

carbon dioxide usage in the ferrous metallurgy process is expected to be more than 100 kg per ton steel.

Round  3

Reviewer 1 Report

In my opinion, there are just two technical changes necessary:

Section 2.3:

What does "average density of inclusion" mean? Is it the number of inclusions per analyzed area?

Section 3.2:

It is not correct to state that CO2 injection into the blast furnace an reduce emission of CO2. It is now stated correctly that the reductant rate will be increased because of CO2 injection. This inevitably means that CO2 emissions from ironmaking will increase.

Author Response

Dear Peer reviewer

I have revised the manuscript according to your suggestions, the detailed responses are shown as follows.

 Section 2.3:

What does "average density of inclusion" mean? Is it the number of inclusions per analyzed area?

Response:

Thank you, “the average number of inclusions per analyzed area decreased” expresses more accurately.

Section 3.2:

It is not correct to state that CO2 injection into the blast furnace an reduce emission of CO2. It is now stated correctly that the reductant rate will be increased because of CO2 injection. This inevitably means that CO2 emissions from ironmaking will increase.

Response:

1)    First reaction

All the O2 and CO2 blown in BF react with burning carbon, and generate into CO firstly in area of blast furnace tuyere.

Because that reaction of CO2 is endothermic, and reaction of O2 is exothermic, a higher oxygen enrichment can be achieved, but the temperature in blast furnace tuyere must keep the same, because of the refractory material.

So the quantity of heat in first gas is similar, CO content in first gas is higher, when CO2 is blown.

At the same time, O2 and CO2 are blown, rate of air in blast furnace blasting is corresponding reduced. Without of N2, the total volume of first gas is lower.

2)    Second reaction

Part of CO in first gas reacts with Iron ore gradually, and CO2 is generated.

the reaction of CO generate into CO2 is endothermic.

When the height of the iron ore is enough, the second reaction is determined by the quantity of heat in first gas, which are almost the same. the content of CO reacts with the iron keeps same, and the content of CO2 is generated keeps same too.

Because that the total volume of first gas is lower, the total generated CO2 is lower, the CO2 emissions from blast furnace ironmaking decreases too.

When CO2 are blown, CO content in first gas is higher, the active amount of CO keeps same, so the remaining CO increases, the calorific value of blast furnace gas and the total amount of CO all increase obviously.

 So the reductant rate of the Blast Furnace is increased, and the production efficiency of BF will be improved, but direct CO2 emissions from ironmaking will decrease. The CO content in BF gas can substitute for more other fuels in other application areas, or be used as chemical raw material, such as for Methanol (CH3OH) production.